# Biological Hallmarks and Emerging Strategies to Target STAT3 Signaling in Multiple Myeloma

**DOI:** 10.3390/cells11060941

**Published:** 2022-03-10

**Authors:** Jianbiao Zhou, Wee-Joo Chng

**Affiliations:** 1Cancer Science Institute of Singapore, National University of Singapore, 14 Medical Drive, Singapore 117599, Singapore; mdccwj@nus.edu.sg; 2Department of Medicine, Yong Loo Lin School of Medicine, National University of Singapore, Singapore 117597, Singapore; 3Department of Hematology-Oncology, National University Cancer Institute of Singapore (NCIS), The National University Health System (NUHS), 1E, Kent Ridge Road, Singapore 119228, Singapore

**Keywords:** multiple myeloma, signal transducer and activator of transcription 3 (STAT3), hallmarks of cancer, targeted therapy, JAK inhibitor

## Abstract

Multiple myeloma (MM) is the second most common hematological malignancy, characterized by an abnormal accumulation of plasma cells in the bone marrow. Signal transducer and activator of transcription 3 (STAT3) is a cytoplasmic transcription factor that modulates the transcription of multiple genes to regulate various principal biological functions, for example, cell proliferation and survival, stemness, inflammation and immune responses. Aberrant STAT3 activation has been identified as a key driver of tumorigenesis in many types of cancers, including MM. Herein, we summarize the current evidence for the role of STAT3 in affecting cancer hallmark traits by: (1) sustaining MM cell survival and proliferation, (2) regulating tumor microenvironment, (3) inducing immunosuppression. We also provide an update of different strategies for targeting STAT3 in MM with special emphasis on JAK inhibitors that are currently undergoing clinical trials. Finally, we discuss the challenges and future direction of understanding STAT3 signaling in MM biology and the clinical development of STAT3 inhibitors.

## 1. Introduction

Multiple myeloma (MM) is a plasma cell neoplasm that results from the clonal expansion of an immunoglobulin-secreting terminally differentiated B cell [1]. As a progressive disease, MM is frequently preceded by premalignant asymptomatic conditions, monoclonal gammopathy of undetermined significance (MGUS) or smoldering myeloma (SMM) [2]. At the advanced stage, MM is characterized by tissue damage that involves bone, kidney, and other organs [3]. During the last two decades, significant improvements have been made in the understanding of myeloma biology and the development of novel treatment strategies, such as proteasome inhibitors, immunomodulatory drugs, and monoclonal antibodies targeting CD38 or B-cell maturation antigen (BCMA)-directed chimeric antigen receptor (CAR) T-cell therapy [4,5,6]. In general, these advances have led to better response rates and improved outcomes. However, MM remains an incurable disease with a five-year survival rate of around 54% [7]. Furthermore, long-term remissions are rare. The challenges in treating MM partially arise from the remarkable heterogeneity in the genetic and epigenetic abnormalities of this disease [8,9]. Cytogenetic analysis has shown that about 90% of patients have recurrent chromosomal abnormalities (e.g., gain or loss of chromosome, deletions, translocations), further complicated by random genetic defects and clonal evolution in an individual patient [10,11,12]. These accumulated abnormalities throughout the disease course have aberrantly activated or inhibited numerous important signaling pathways, thus leading to aggressive phenotypes, worsening prognosis and nullifying treatment response [13,14]. Therefore, the exploration and identification of some common signaling pathways shared by a significant percentage of the patient pool for the management of MM are urgently needed. Hence, in the current review, we aim to discuss the emerging role of the signal transducer and activator of transcription 3 (STAT3) in the biological hallmarks of MM and explore its potential as an effective therapeutic target for managing this disease.

## 2. STAT Signaling and Functions

The family of signal transducer and activator of transcription (STAT) proteins are latent transcription factors which are activated when extracellular ligands bind to surface receptors. Janus tyrosine kinases (JAKs)-STAT pathway was originally discovered in the context of interleukin-mediated inflammatory response [15]. There are seven members of the STAT family, including STAT1, STAT2, STAT3, STAT4, STAT5A, STAT5B, and STAT6. The family of STAT shares a common molecular topology which is organized into specific functional modular domains. STAT proteins are composed of an N-terminal domain (N-term), a coiled-coil domain (CCD), a DNA binding domain (DBD), a linker domain (LD), an SRC homology 2 domain (SH2), and a C-terminus named the trans-activation domain (TAD). From the function aspect, the N-domain is involved in the formation of STAT tetramers bound to adjacent DNA sites. The CCD, containing several alpha-helices in a ropelike structure, facilitates nuclear export/import via a nuclear localization signal (NLS) motif. The DBD has a conserved immunoglobin-like structure that specifically binds to distinct DNA target sequences, a palindromic TTCN3-4GAA motif (consensus STAT motif). The adjacent LD provides structural support not only for the DBD binding to target DNA, but also for the formation of transcriptional complexes [16]. The SH2 domain is able to bind the phosphotyrosine (pTyr, pY) of activated STAT proteins, promoting dimerization of two STAT monomers. The C-terminal TAD interacts with transcriptional coactivators such as CBP/p300 or MCM5 [17,18] and the phosphorylation of a serine residue (pSer, pS) as the TAD can further increase transcriptional activation [18].

In their updated model, Hanahan and Weinberg proposed eight “hallmarks of cancer” that all malignancies must achieve, including sustaining proliferative signaling, evading growth suppressors, resisting cell death, enabling replicative immortality, inducing angiogenesis, activating invasion and metastasis, reprogramming of energy metabolism and evading immune destruction [19]. The STAT proteins play a pivotal role in diverse biological functions, including cell proliferation, cell survival, angiogenesis, apoptosis, and inflammation. Accordingly, aberrant activation of the STAT pathway has been widely identified in a number of human diseases, particularly immune deficiencies, inflammatory diseases, cancers and other proliferative disorders. Compared to the other four members, STAT3, STAT5A and STAT5B have been implicated in a majority of solid tumors and hematological malignancies. They contribute to cancer cell survival, proliferation and resistance to apoptosis, STAT3 has been demonstrated to have additional functions in the tumor microenvironment and immune escape of cancer cells.

## 3. STAT3 Sustains MM Cell Survival and Proliferation

The majority of CD138+ primary MM cells express constitutively active STAT3 [20]. It is well known that a number of upstream cytokine and growth factors can activate the STATs signaling pathway. Among them, IL-6, the most studied cytokine for hyperactivation of the JAK-STAT3 pathway, is crucial for the survival and proliferation of MM cells. Gene expression profiling (GEP) demonstrated that the regulation of most IL-6 target genes required the activation of STAT3 [21,22]. These two studies perform GEP experiments on different human MM cell lines upon IL-6 stimulation. We extracted 15 common upregulated genes upon activation of IL6-STAT3 pathway shared by these two datasets (Figure 1). This MM-STAT3 signature includes B-cell CLL/lymphoma protein family (BCL3 and BCL6), JUN oncogenes (JUN and JUNB), interferon regulatory factor (IRF1 and IRF9), Bcl2 family (MCL1), protein tyrosine phosphatase type IVA, member 3 (PTP4A3, also named as PRL-3), etc. BCL6 variant 2 is absent in normal plasma cells, but upregulated by STAT3, which is bound on the promoter region of the BCL6 gene in IL-6-dependent MM cell lines. Importantly, silencing BCL6 decreases MM cell survival and proliferation [22]. Chromosome 1q21 amplification confers poor prognosis in multiple myeloma. Our group has identified that IL6-R (IL6 membrane receptor) and ADAR1 (RNA editing enzyme) are critical genes located within the minimally amplified 1q21 region. ADAR1-P150 functions as a direct transcriptional target for STAT3 and ADAR1 forms a constitutive feed-forward loop with STAT3 to stimulate oncogenic transcription factors for pro-survival genes, including MCL1, BCL2 and ADAR1-P150 itself [23,24].

Protein tyrosine phosphatases (PTPs) which remove a phosphate group from a tyrosine residue, have gained prominence with emerging evidence implicating them in a variety of diseases including diabetes, obesity, autoimmune disease and cancers [25]. PRL-3 (encoded by PTP4A3 gene) is the third member of the phosphatase of regenerating liver (PRL) family [26,27]. Bert Vogelstein’s group first identified that PRL-3 was overexpressed in metastatic colorectal cancer [28]. Since then, there is increasing evidence that support an oncogenic role of PRL-3 in tumorigenesis, progression and metastasis across different cancers [26,27,29,30,31]. PRL-3 is overexpressed in MGUS, SMM, and MM, compared to PCs from healthy persons. Furthermore, a high level of PRL-3 defines a novel cluster of MM patients in GEP analysis and predicts poor survival myeloma patients [32,33]. Studies from our laboratory and others demonstrate the existence of a positive feedback regulatory loop between STAT3 and PRL-3 in MM [34,35]. IL-6 promotes STAT3-dependent transcriptional upregulation of PRL-3, while high PRL-3 in turn re-phosphorylates STAT3 and aberrantly hyperactivates STAT3 target genes, including c-MYC, MCL1, BCL2 and BCL-xL [34,35]. Thus, this regulatory pathway is essential for MM survival, proliferation and resistance to drug treatment.

In addition to IL-6, a list of other cytokines and molecules have been described to be STAT3 upstream activators, such as heme oxygenase-1 (HO-1) [36], B7-H3 (CD276) [37], interferon (IFN)-lambda1 (IFN-λ1/IL-29) [38].

## 4. STAT3 Regulates Tumor Microenvironment

In the original “hallmarks of cancer” model, inducing angiogenesis and activating invasion and metastasis all intertwined with tumor microenvironment, which has been traditionally considered in the context of solid tumor [19]. Similarly, in hematological malignancies (multiple myeloma herein), bone marrow (BM) milieu not only promotes tumor cell survival and progression, but also furnishes a sanctuary to protect from cytotoxic drugs and resist apoptosis. The BM microenvironment is a complex ecosystem, comprising hemopoietic cells and non-hematopoietic cells (for example, stromal cells, endothelial cells, osteoblasts, immune cells, etc.), as well as the non-cellular components (for example, extracellular matrix (ECM), chemokines, cytokines and growth factors). The BM microenvironment (niche) has recently been recognized as essential for myeloma development and chemoresistance by constitutive activation of STAT3 signaling.

BM stromal cells (BMSCs) secrete IL-6, resulting in a STAT3-dependent overexpression of MCL1, a member of the prosurvival BCL2 family, promoting MM survival and resistance to BCL2/BCL-xL inhibitor ABT-737 [39]. Inhibition of the JAK-STAT3 downstream ERK pathway could reverse this drug resistance phenotype [39]. Consistently, independent studies demonstrated that in the BM niche, there is other prosurvival signaling in addition to the IL-6/JAK/STAT3 pathway [40,41]. Cotargeting STAT3 and MEK/ERK or RAS/mitogen-activated protein kinase (MAPK) pathway produces synergistic effects and leads to the robust induction of apoptosis in MM cells grown with BMSCs [40,41]. In addition, BCL6 expression in MM cells is increased by the stimuli from BM niche via IL-6/JAK/STAT3 and canonical nuclear factor-kappaB (NFκB) pathways [42]. Direct targeting BCL6 directly or upstream STAT3 and NFκB cascades inhibits the survival of MM cells even in their BM niche [42]. Daratumumab (DARA), an anti-CD38 monoclonal antibody (MoAb) has shown potent therapeutic effects for both newly diagnosed and relapsed multiple myeloma (MM) [43]. However, the coculture of MM cells with BM stromal cells or IL-6 activates the STAT3 pathway, leading to downregulation of CD38 expression on MM cells, which in turn renders resistance to DARA [44]. Taken together, STAT3 is hyperactivated in both MM cells and BM milieu. Combined targeting of the MM ecosystem is required to achieve desired and sustained clinical benefits.

Integrins are heterodimeric receptors, consisting of 18 α and 8 β subunits which form 24 different non-covalent integrin [45]. There are two-way interactions and signaling interplays between integrins and ECM, cell surface molecules or soluble factors [46]. Thus, integrins are important for transmitting signals between MM cells and their BM milieu. The binding of BM component fibronectin (FN) to β1 integrins on MM cells results in cell adhesion-mediated drug resistance (CAM-DR) [47]. Mechanistically, concomitant exposure of MM cells to IL-6 and FN adhesion bring forth a remarkable increase in STAT3 activity through a novel association between STAT3 and gp130. Proline-rich tyrosine kinase 2 (PYK2) is a critical downstream effector of this pathway within the context of microenvironmental cues [47,48]. Overall, this β1 integrin/gp130/STAT3 signaling provides MM cells with additional survival advantages in their bone marrow milieu.

## 5. STAT3 Induces Immunosuppression

The human immune system can be divided into two different, but integrated compartments: the innate (general) and the adaptive (specialized) immunity. The innate immune system serves as the first line of defense that consists of natural killer (NK) cells, mast cells, eosinophils, basophils, macrophages, neutrophils, dendritic cells (DCs) and humoral factors (cytokines and complement). An important function of innate and adaptive immunity in the defense against cancer is surveillance and identification of foreign or “non-self” substances. However, cancer cells develop diverse mechanisms to avoid immune recognition, produce immune suppressive cytokines and an immunosuppressive tumor microenvironment, thus evading immune surveillance.

A large body of evidence supports that STAT3 play a pivotal role in immunosuppression exerted by MM cells and BM microenvironment. DCs are a group of heterogeneous immune cells that converge innate and adaptive immunity. DCs are professional antigen-presenting cells that capture, process, and present antigens to adaptive immune cells and mediate their polarization into effector cells. Tumor-derived factors induce the constitutive activation of JAK/STAT3 and prevent the differentiation of immature myeloid cells into mature dendritic cells, resulting in a decreased number of mature DCs [49]. Pretreatment of myeloma cells with STAT3 inhibitor, JSI-124 and bortezomib can reverse compromised DC function, generating potent myeloma-specific cytotoxic T lymphocytes (CTLs) via the inhibition of HSP90 and STAT3 activity [50]. When compared to healthy controls, the function of mature, high-density neutrophils (HDNs) in patients with MM and MGUS is impaired, showing reduced phagocytic activity and oxidative burst [51]. This impaired function is caused by upregulation of the inducible FcγRI (CD64) and a downregulation of the constitutive FcγRIIIa (CD16), leading to the aberrant activation of STAT3 [51].

Exosomes are small membrane vesicles of endocytic origin, size (diameter) ranging from 30–150 nm [52]. They are secreted by hematopoietic cells or non-hematopoietic cells into extracellular space. Emerging evidence demonstrates that exosomes are not only important mediators of tumorigenesis, but also serve prognostic and diagnostic markers, as well as having a therapeutic function in cancer [53]. MM exosomes stimulate the STAT3 pathway in endothelial and BM stromal cells, thus enhancing angiogenesis and endothelial cell growth [54]. Furthermore, MM exosomes increase the immunosuppressive capacity of myeloid-derived suppressor cells (MDSCs) through the activation of the STAT3 pathway, facilitating MM progression [54].

Activation of STAT3 has been found to inhibit NK cell-mediated innate immunosurveillance. Targeting STAT3 with small molecule inhibitor or siRNA increases NK degranulation and IFN-γ production in a TGF-β1-independent manner. The natural killer group 2, member D receptor (NKG2D) is an activating cell surface receptor that is predominantly expressed on cytotoxic immune cells. In MM cells, STAT3 activated by glycogen synthase kinase-3 (GSK3), a pleiotropic serine-threonine kinase, binds to the promoter of MHC class I chain-related protein A (MICA), leading to the upregulation of MICA. MICA is a ligand for NKG2D. MM cells secrete soluble MICA, which inhibits NK cell degranulation and decreases the ability of myeloma cells to trigger NK cell-mediated cytotoxicity. Thus, STAT3 has a novel role in the evasion of NK cell immunosurveillance by modulating MICA expression in MM cells.

## 6. Targeting STAT3: Killing Many Birds with One Stone

As summarized above, constitutive activation of STAT3 plays a critical role in several aspects of MM pathogenesis, including cell survival, proliferation, resistance to apoptosis, BM microenvironment and immune evasion, So, targeting STAT3 can achieve multiple benefits simultaneously (Figure 2). Different strategies for targeting STAT3 have been developed as novel treatment options for MM. Direct inhibition of STAT3 with SH2 domain inhibitor OPB51602 recorded significant toxicity and no response in a phase I clinical trial of 20 RR-MM patients [55]. Here, we summarize the studies targeting upstream regulators of STAT3, which have been shown to be clinically promising at the current stage.

IL-6 antagonist. The IL-6/JAK/STAT3 regulatory axis interleukin-6 (IL-6) plays a central role in the pathogenesis of several cancers, including MM. Sant7 is a monoclonal antibody against IL-6 receptor (IL-6), a super antagonist [56,57]. Sant7 has higher affinity for gp80 than IL-6, but lacks binding capacity to the gp130 receptor signaling subunit, thus potently inhibiting IL-6/STAT3 cascades [56,57]. The apoptotic effect on MM cells treated with Sant7 correlated with its affinity for IL-6, degree of gp130 binding impairment, and efficiency to inhibit STAT3 activity [56]. A combination therapy of Sant7 with dexamethasone induced synergistic anti-myeloma in vitro and in mouse xenograft models [58,59,60,61], and overcame BM stromal cell-mediated drug resistance [40,62].

JAK inhibitors. Ruxolitinib (INCB018424 or INC424) is a potent and selective JAK1 and JAK2 inhibitor by competitively inhibiting the ATP-binding catalytic site on JAK1 and JAK2. Ruxolitinib has been approved for the treatment of myelofibrosis, polycythemia vera, and graft-versus-host disease. A number of preclinical studies have tested the combination of ruxolitinib with established anti-myeloma agents, as well as with some novel therapeutics in the preclinical setting. Cotreatment of ruxolitinib with bortezomib, lenalidomide and dexamethasone, inhibits the proliferation of the MM cell lines and induces cell cycle arrest [63], overcomes resistance of myeloma to lenalidomide by suppression of macrophage-2 (M2) polarization [64]. In addition to M2 polarization, inhibition of STAT3 signaling by ruxolitinib leads to a decreased expression of programmed cell death ligand-1 (PD-L1) and increases the antitumor effects of T cells in MM [65].

Ruxolitinib upregulates CD38 expression via inhibition of STAT3 phosphorylation, thus enhances the DARA-mediated antibody-dependent cellular cytotoxicity (ADCC) against MM cell lines and primary CD138+ myeloma cells derived from MM patients [44]. In clinic, a phase I trial of ruxolitinib, lenalidomide, and steroids observed a 46% clinical benefit rate and a 38% overall response rate in 28 relapsed/refractory (RR)-MM patients who had received at least six prior treatments before enrollment [66]. These data demonstrated that the JAK inhibitor, ruxolitinib, could restore the sensitivity of lenalidomide and steroids for patients with RRMM [66]. Sporadic case patients with co-existing myeloproliferative neoplasms (MPNs) and plasma cell disorders (PCDs) who received ruxolitinib showed improvement in the myeloma biomarkers, suggesting an anti-myeloma effect from ruxolitinib treatment [67]. Other selective JAK inhibitors, including INCB20, AZD1480, INCB16562 CYT387 and multikinase inhibitor, AT9283, have shown potent anti-myeloma effects in preclinical studies, too [68,69,70,71,72]. Taken together, these results represent that inhibition of the JAK-STAT3 pathway is a promising novel therapeutic approach for treating multiple myeloma.

## 7. Conclusions

The experimental and clinical evidence reviewed herein suggests a key role for STAT3 in favoring several “hallmark of cancer” traits in the pathogenesis of multiple myeloma. Most importantly, the constitutive activation of STAT3 signaling in both MM cells and BM microenvironment, builds a positive feedback loop between MM cells and their niche for their survival and escaping immunosurveillance. Thus, STAT3 is instrumental for the emergence of drug resistance and the survival of these resistant clones is responsible for refractory/relapsed cases in MM patients.

Despite the scientific robustness, most experimental evidence on STAT3 was obtained from human MM cell lines in an in vitro setting and to a lesser extent from primary MM cells or in in vivo models. Thereby, we face a major challenge in mapping the dynamic MM-niche interaction within living systems, although we know, in considerable detail, about the functions of STAT3 in MM biology. It is advisable to attain better in-depth understanding of the aberrant STAT3 signaling network in MM cells in their native BM niche. Especially, the advances in high-resolution imaging together with computational analysis can provide novel biological insight by mapping preferential spatial affinity and dynamics of MM cells for contacting the vascular niche in their native environment.

Because of its multifarious role in tumorigenesis, targeting STAT3 signaling is emerging as an attractive approach to treat MM. Several strategies have been developed to inhibit STAT3 signaling at multiple levels. With the aspect of clinical application, the majority of these approaches are still at an early stage and only supported by preclinical data and most of compounds are not suitable for further clinical development due to severe toxicity or poor pharmacokinetic properties. Currently, the JAK inhibitor ruxolitinib is in its most advanced clinical trial stage. Results from the combination regimen of ruxolitinib, dexamethasone, and lenalidomide for patients with heavily pretreated RR-MM demonstrate evident clinical activity and well-tolerated side effects. Early results from the two-drug combination of ruxolitinib and methylprednisolone for RR-MM patients are also encouraging (NCT03110822). Given these findings, we expect ruxolitinib to be expanded into large trials and other JAK inhibitors to be evaluated in clinic. 

## Figures and Tables

**Figure 1 cells-11-00941-f001:**
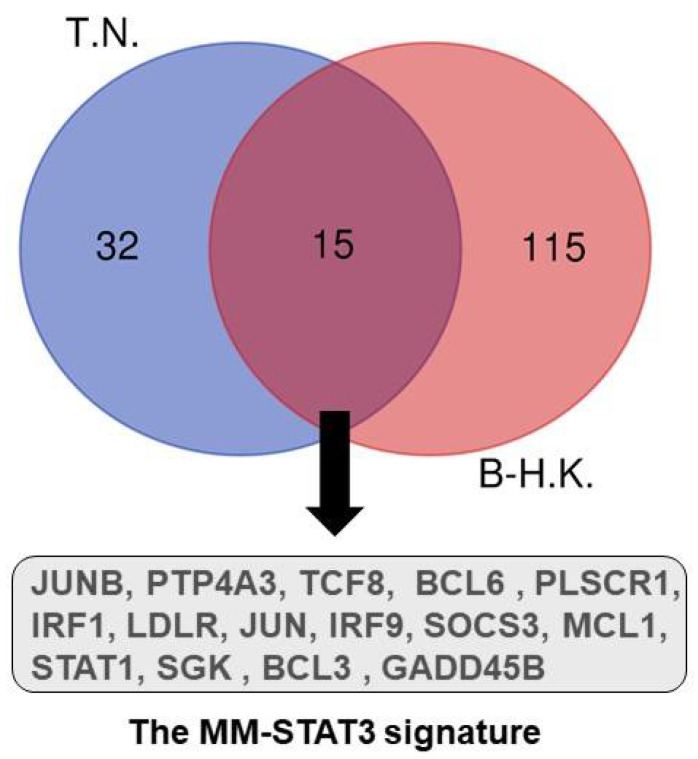
Venn diagrams of genes upregulated in MM cell lines after additional IL-6 stimulation. The MM-STAT3 signature contains 15 upregulated genes shared by studies of Brocke-Heidrich K, et al. (B-H.K., ref. [21]) and Tsuyama N (T.N., ref. [22]).

**Figure 2 cells-11-00941-f002:**
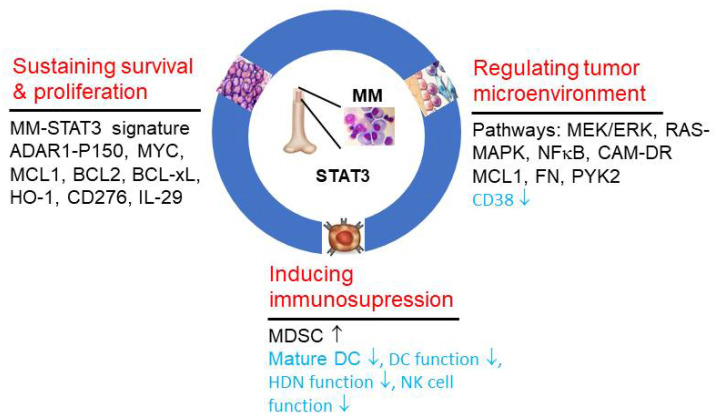
STAT3 regulates the biological hallmarks and enabling characteristics of multiple myeloma. The key mechanisms of action undertaken by STAT3 in facilitating the hallmarks: (1) sustaining MM cell survival and proliferation; (2) regulating tumor microenvironment (bone marrow niche); (3) inducing immunosuppression, and characteristic pathways and essential genes are listed. Genes or pathways in black font represent overexpression or activation, while gene and cell type and function with down arrows in blue font denote decreased expression, or number or function.

## Data Availability

Not applicable.

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
