# Peer review of "Biological Hallmarks and Emerging Strategies to Target STAT3 Signaling in Multiple Myeloma"

_cells, 2022, doi:10.3390/cells11060941_

Round 1

Reviewer 1 Report

This is an interesting review summarizing some evidence for the role of STAT3 in 1) sustaining MM cell survival and proliferation,  2) regulating tumor microenvironment and 3) inducing immunosuppression. In addition, the authors discuss different strategies for targeting this molecule in MM .

I think that a diagram depicting the three points they described as well as the STAT3 targeting strategies is required, as it will be helpful for the readers.

The authors should also correct lines 16, 104,166,180 and 293.

Reviewer 2 Report

In this review, Zhou and Chng summarized the role of STAT3 signaling pathways in multiple myeloma and the potential target of STAT for myeloma treatment. Simulation of IL-6/ Jak/STAT signaling is a common mechanism in cancers. Chng et al had published another review of JAK and lymphoid cancers in 2021. However, the novelty of this manuscript in myeloma field is significant. Regarding the well-organized structure and comprehensive references, I agree to accept this article with following questions and comments:

  1. The first sentence of the abstract mentioned“ Multiple myeloma (MM) is the most common hematological malignancy characterized by…. Actually MM is the second most common hematological malignancy. Please clarify it.
  2. In the part STAT3 regulates tumor microenvironment. Authors mentioned “…non-hematopoietic cells (for example, stromal cells,endothelial cells, osteoclasts, osteoblasts, immune cells, etc),” Osteoclasts are derived from myeloid progenitors therefore belong to hematopoietic cells. Please fix it.
  3. In the part STAT3 induces immunosuppression, please simplify the explanation of the innate (general) and the adaptive (specialized) immunity, which is quite familiar to readers.
